# Early Bolt Loosening Detection Method Based on Digital Image Correlation

**DOI:** 10.3390/s24165397

**Published:** 2024-08-21

**Authors:** Yinyin Li, Yusen Wu, Kang Gao, Huiyuan Yang

**Affiliations:** 1School of Mechanical Engineering, Yangtze University, Jingzhou 434023, China; 2022720615@yangtzeu.edu.cn (Y.W.); 2023720659@yangtzeu.edu.cn (K.G.); 2College Vehicle and Traffic Engineering, Henan University of Science and Technology, Luoyang 471000, China

**Keywords:** digital image correlation, early bolt loosening assessment, average strain, axial load

## Abstract

Bolt loosening can significantly impact the accuracy, stability, and safety of equipment. The detection of bolt loosening in a timely manner is crucial for ensuring the safety, reliability, performance, and service life of equipment, structures, and systems. Various methods exist for detecting bolt loosening, such as strain gauges and ultrasonic waves. However, these technologies have some limitations that impede their widespread application. In this paper, for the high-pressure pipe manifolds that may experience leakage accidents due to the loosening of bolts, an early bolt loosening detection method based on digital image correlation is proposed. Initially, a model is established through tensile tests to relate the average strain on the side of the bolt head to the axial force. Subsequently, an industrial camera captures images of bolts with random speckles under operational conditions. Using digital image correlation technology, the average strain in a specific region on the side of the bolt head is calculated. By integrating the average strain into the established relationship model between the average strain and axial force, the axial force of the bolt under operational conditions can be predicted, enabling the early assessment of bolt loosening. The findings show that the average strain on the side of the bolt head increases proportionally with the axial force, indicating a strong linear relationship. This method enables accurate prediction of the bolt’s axial force, offering a new approach for identifying the early loosening of bolts in high-pressure manifolds and monitoring structural health.

## 1. Introduction

Bolts are the most common fasteners in industrial production, widely used in digital electronics, aerospace engineering, bridges, vehicles, oil and gas pipeline networks, and many other areas of industry because of their good disassembly, sealing, fatigue resistance, and other characteristics. According to statistics, threaded connections account for around 68% of all connections, making bolts the fundamental link in the manufacturing industry [1]. Bolts are exposed to various conditions during operation, such as various temperatures, vibration, and alternating equipment forces, which may cause them to loosen. This can have a severe negative impact on the safety and stability of structures, potentially leading to equipment failure, structural damage, and accidents, resulting in incalculable losses and serious consequences for both enterprises and society. Research indicates that 70 to 90% of mechanical equipment failure modes are directly related to bolt failure [1]. Furthermore, statistics from the UK’s UHOOA show that nearly 81% of 100 flange leakage accidents were caused by preload issues. Therefore, the detection of bolt loosening in a timely manner during operation is crucial for ensuring production safety.

Researchers have developed various methods for detecting bolt loosening, such as those based on acoustics [2,3,4], strain gauges [5,6], electromechanical impedance [7,8,9], and piezo-ceramic sensors [10,11,12]. Although these have shown some potential in detecting bolt loosening, they also have certain limitations. For instance, they require complex equipment to obtain precise measurement results and may face difficulties in practical applications [13]. Due to the development of computer hardware technology, image processing technology, and related algorithms [14,15,16], the speed of image processing by digital image technology is accelerating, and it has application potential in many fields. Similarly, in the detection of bolt loosening, the research of digital-image-related technology has also made great progress. Compared to traditional detection methods, digital image technology offers simplicity in operation, high precision, and simple equipment requirements, and it enables non-contact detection. Currently, most research on bolt loosening detection methods based on digital image technology focuses on quantifying the bolt loosening angle. For example, Kong [17] used an image registration method to estimate the bolt loosening angle, while Deng [18] employed a deep learning algorithm and geometric imaging theory to detect the loosening angle of mark bolted joints. Xie used a digital image correlation method to quantitatively assess the relationship between bolt preload loss and the bolt loosening angle [19]. Additionally, there are approaches that utilize the Hough transform algorithm to segment the bolt edge and detect the loosening angle [20].

However, in some working environments, despite no apparent change in the angle of the bolt head, loosening accidents may still occur. For instance, in oil and gas extraction, a high-pressure pipe manifold may cause a leakage accident due to the loosening of bolts, as shown in Figure 1a. In this case, the high-pressure pipe manifold does not have a significant deflection angle. This accident is caused by the fact that the machined surface is not an ideal smooth plane. From a microscopic perspective, it is actually composed of countless small convex peaks, and the rough surface contour is composed of a series of “peaks” and “valleys”, as shown in Figure 1b,c [21]. In connecting components, the stiffness of the sealing part is significantly lower than that of the sealed part. When there is a restricted fit and medium pressure, the softer sealing surface undergoes significant elastic–plastic deformation to fill the leakage pores. When bolts are used under high-pressure conditions, they may be affected by various factors such as pressure fluctuations and vibrations, which may lead to the expansion of leakage pores. In such cases, even if there is no significant change in the bolt angle, early loosening may occur, leading to leakage accidents. This early loosening is difficult to detect solely by measuring the loosening angle of the bolt.

The digital image correlation (DIC) technique is a non-contact image measurement technology based on computer vision. It is widely used for measuring the full-field shape, deformation, and motion, enabling the precise measurement of displacement and strain information on object surfaces [23,24]. This technology has been applied in numerous industries. Farahani et al. used the DIC method to calculate the force/displacement response of AA6061-T6 plate specimens under static tension, effectively predicting the damage and failure phenomena of the material [25]. Yuan et al. used the DIC technique to observe the full-field deformation of seawater-mixed aluminate cement concrete during compression under freshwater/seawater curing conditions and to obtain the displacement and strain fields of the concrete cubes [26]. Wu et al. proposed a DIC method for thermal flow excitation, which effectively detects internal cracks in 304 stainless-steel material by heating the specimen with thermal excitation and measuring the strain field using the DIC method [27]. Li et al. successfully utilized the DIC method to investigate the tensile mechanical properties of rigid insulation tile materials under high temperatures ranging from 700 to 1000 °C [28]. These studies have demonstrated that the DIC method is reliable and accurate at both room and high temperatures.

Due to the good linear relationship between the deformation of the bolt head and the axial force acting on the bolt [29], this study addresses the possibility of leakage incidents at high-pressure manifolds due to the loosening of bolts and proposes an early bolt loosening detection method based on DIC technology. By capturing and processing digital images, accurate strain information on the side of the bolt head is obtained, enabling the calculation of the tensile load on the bolt to determine whether there are signs of loosening. The rest of this paper is structured as follows: Section 2 elaborates on the principles of DIC technology, discusses potential interference factors, and suggests corresponding mitigation strategies. The experimental setup and method are detailed in Section 3. Section 4 presents the experimental results and accompanying discussions. Finally, Section 5 summarizes the key findings, identifies the limitations of the current method, and outlines potential directions for future improvements. This method can easily and accurately realize the prediction of bolt axial force, study the performance changes of high-pressure pipe manifold flange bolts during long-term operation, assess their stability and reliability, and provide a new method for the early loosening of high-pressure pipe manifold bolts and structural health detection.

## 2. Theoretical Background and Signal Processing

### 2.1. Principles of DIC Technology

The principle of DIC technology [30] involves comparing the feature information in digital images of an object’s surface before and after deformation to calculate physical quantities such as displacements and strains. The fundamental approach is to capture speckle images of the object’s surface in both its undeformed and deformed states. A small section of the undeformed image is designated as the reference sub-region, and the corresponding section in the deformed image that matches the reference sub-region is identified as the deformed sub-region, as shown in Figure 2a,b. By establishing the correspondence between these two sub-regions, the deformation magnitude can be extracted. The difference between the reference sub-region and the deformed sub-region has a displacement component in the position and a strain component in the shape, which turns the deformation measurement problem into a computationally tractable digital correlation process [31].

In the process of searching for deformed sub-regions, a global and quantitative function is needed to express the relationship between the reference sub-region and the deformed sub-region, which can be expressed by the correlation coefficient. The correlation coefficient can be defined as [32]
(1)$$A(X)=\frac{\sum f(x,y)g(x_{1},y_{1})}{\sqrt{\sum f^{2}(x,y)\sum g^{2}(x_{1},y_{1})}}$$
where x and $x_{1}$ represent the horizontal coordinates before and after the deformation, respectively. Similarly, y and $y_{1}$ represent the vertical coordinates before and after the deformation, respectively. $X=(u,v,\frac{\delta u}{\delta x},\frac{\delta u}{\delta y},\frac{\delta v}{\delta x},\frac{\delta v}{\delta y})$ represents the variables to be solved, and *u* and *v* represent the displacements in the *x* and *y* directions of the center point of the deformation region, respectively. Additionally, f(x,y) represents the grayscale of the reference sub-region, while $g(x_{1},y_{1})$ represents that of the deformation sub-region.

Considering the simplest case, where the deformed sub-region has only displacement but no deformation, the relationship between the coordinate changes of a point G in the image before and after deformation can be expressed by the following equation:(2)$$\left\{\begin{array}{c} x_{1}=x+u\\ y_{1}=y+v \end{array}\right.$$

When considering the rotation, telescopic deformation, and shear deformation in the deformation sub-region, the relationship between the coordinate changes before and after the deformation of a point G in the image can be expressed as follows.
(3)$$\left\{\begin{array}{c} x_{1}=x+u+\frac{\delta u}{\delta x}\Delta x+\frac{\delta u}{\delta y}\Delta y\\ y_{1}=y+v+\frac{\delta v}{\delta x}\Delta x+\frac{\delta v}{\delta y}\Delta y \end{array}\right.$$

When the correlation coefficient is equal to 1 or 0, the two sub-regions are perfectly correlated or uncorrelated, respectively [31].

The most basic way to calculate strain is by performing differential operations on the displacement data. However, this method can amplify signal noise or errors, resulting in inaccuracies in the real strain value. A more effective approach is the local least-squares method for the displacement field, which helps to suppress noise in strain calculations with greater precision. The Cauchy strain components derived from the displacement field using the local least-squares method are [33]
(4)$$\left\{\begin{array}{c} \varepsilon_{x}=\frac{\partial u}{\partial x}\\ \varepsilon_{y}=\frac{\partial v}{\partial y} \end{array}\right.$$

### 2.2. Analysis and Solutions of Influencing Factors in Strain Detection Based on DIC Method

When employing the DIC technique for measuring strains on the side of the bolt head, several factors can impede its accuracy. The primary factors of interference include the following three aspects.

#### 2.2.1. Correlation Loss

DIC techniques, especially phase correlation methods, rely on the textural features of the image for similarity assessments. It is essential that the sub-regions of the image exhibit rich textures; otherwise, correlation loss may occur if the sub-regions are absent or if their textures vary significantly, thereby impacting the accuracy of the measurement results [34]. This study aims to address correlation loss through two primary strategies: (1) Enhancing the texture of the object under examination, which can be achieved by applying custom scattering. As only a few specimens possess the scattering properties necessary for digital image computations, it is often essential to artificially introduce scattering using tools such as a spray gun to meet the computational requirements. To ensure the accuracy and reliability of subsequent computations, this scattering must possess specific characteristics, including an appropriate size, a high level of random dispersion, an optimal density, and matte attributes. It is crucial to limit the size of the scattering to a range of 3 to 5 pixels to avoid distortion caused by excessively small sizes or resolution reduction in displacement measurements due to overly large sizes. Moreover, the scattering distribution should be suitably random to ensure uniqueness within each region. Maintaining a scattering density of around 50% is vital to ensure an adequate number of feature points for DIC computations and to prevent image overcrowding [35]. Lastly, using matte paint for scattering helps mitigate adverse effects on image quality resulting from specular reflections. (2) Reasonable division of the image sub-region size. Dividing too large a sub-region makes the computation inefficient, whereas too small a sub-region would cause vector correlation. It is appropriate to take a sub-region that can observe 33 scattering sizes.

#### 2.2.2. Imaging Noise

Practical applications, as opposed to laboratory environments, will encounter various interfering factors, such as noise. In the presence of noise interference, image quality is often degraded, significantly impacting measurement results. The influence of noise on image correlation is illustrated in Figure 3.

Images with more noise need to be preprocessed before image processing to improve the quality of the picture. The main preprocessing processes include the following:

(1) Grayscale conversion: The RGB values of each pixel are converted to a single grayscale value, as shown in Figure 4a,b. This can be achieved through various methods, such as averaging, the maximum and minimum value methods, and the weighted average method. In this study, the latter method is used for preprocessing the images.

(2) Filtering: Images affected by noise interference are eliminated or minimized through digital image processing. Commonly employed filtering techniques comprise median filtering, mean filtering, Gaussian filtering, and others. The selection of the appropriate method depends on the experiment and the type of noise. In this study, Gaussian filtering is utilized with the following formula:(5)$$f_{\sigma }(m,n)=\frac{1}{2\pi \sigma^{2}}e^{\frac{-(m^{2}+n^{2})}{2\sigma^{2}}}$$
where *m* represents the vertical distance between other pixels and the center pixel; *n* represents the horizontal distance between other pixels and the center pixel; and *σ* represents the standard deviation. The parameter *σ* has a significant effect on the filtering effect: when *σ* is small, the filtering effect is weak, but it can retain the image details better; when *σ* is large, the filtering effect is strong, but it may lead to the blurring of the image. *σ* can be selected from 0.5 to 3.5 if it is small, and in this study, we change the value of *σ* and find that the effect is better when *σ* = 1 by observing the processing effect of the actual value. In practice, it is necessary to choose the appropriate value of *σ* according to the specific needs, and the image characteristics also need to use a variety of algorithms for image enhancement processing, such as histogram equalization and prescriptive operations.

#### 2.2.3. Light Intensity

Digital imaging technology is affected by lighting factors during practical application. Under different lighting intensities, when capturing images of the same object, a camera will produce different pixel values. Noise is more likely to be generated in dim lighting conditions [12]. In this case, image quality can be improved by using various image enhancement methods such as DRPN, KinD, the histogram stretching method, and Retinex. Here, we use the histogram stretching method and Retinex to deal with the intensity of the darker image lighting.

The principle of the histogram stretching method is relatively simple: the original minimum value of the grayscale of the image is mapped to 1, the maximum grayscale is mapped to 255, and the intermediate grayscale is mapped in accordance with the ratio, which is divided into linear stretching, three-segmented linear stretching, and nonlinear stretching. Normally, we use linear stretching for image processing, and for the pixel *i* at any point of the image, its grayscale is redefined as
(6)$$h{\left(i\right)}_{new}=255\times \frac{h{\left(i\right)}_{old}-Min}{Max-Min}$$
where *Max* and *Min* represent the maximum and minimum values of the grayscale in the original image, respectively; *h*(*i*)_old_ and *h*(*i*)*_new_* represent the gray value of the pixel before and after the operation, respectively.

The Retinex theory was proposed by Land [36] and others based on the systematic visual perception of the human eye regarding the color and luminance values of the image. This theory decomposes the original image into a light image with grayscale transformation and a reflection image with detailed features of the object. The core idea of the Retinex theory is to retain the feature information of the object in the reflected image and adjust the light component appropriately. The theoretical formula is as follows:(7)S(x,y)=R(x,y)×L(x,y)
where S(x,y) represents the input image, R(x,y) represents the reflection component, and L(x,y) represents the light component.

The basic Retinex theory is usually divided into single- (SSR) and multi-scale Retinex (MSR). MSR deals with the illumination and reflection components at different scales and weights the final result for fusion, which can better preserve image details, especially in the highlights and shadow areas. It is very suitable for addressing noise issues caused by changes in light intensity. In the MSR method, the image is first logarithmically transformed and Gaussian-filtered. Then, weighted fusion and exponential transformation are performed to obtain the enhanced image. The MSR algorithm is formulated as follows:(8)$$r(x,y)=\sum_{k}^{k}w_{k}\{\log I(x,y)-\log[F_{k}(x,y)\times I(x,y)]\}$$
(9)$$F(x,y)=\frac{1}{\sqrt{2\pi }\sigma_{k}}e^{\frac{-(x^{2}+y^{2})}{2\sigma^{2}}}$$
where I(x,y) represents the original low-light image, F(x,y) denotes the Gaussian filter function, and *k* is the number of scales, usually set to 3. When *k* = 1, the MSR degrades to the SSR mode.

### 2.3. Regional Selection

To accurately determine the relationship between the mean strain and axial force on the side of the bolt head, it is recommended to select the horizontal and vertical midlines of a side of the bolt head as the reference. Define a rectangular region with dimensions equal to 50% of the length and width of the bolt head as the region to be measured (referred to as the central region). This central region is illustrated in Figure 5. In this region, the mean strain $\bar{\varepsilon }$ in the y-direction, which is consistent with the axial force on the bolt, is calculated. The strain mentioned later refers to that in this calculation. Compared to calculating the strain field of the bolt head, the computational cost in the central region is smaller, and measuring the average strain in this region provides better stability in practical engineering applications.

## 3. Experiment

### 3.1. Experimental Setup

The proposed method for the early assessment of bolt loosening, based on DIC technology, was validated through experiments. The experiments involved selecting M20, M24, and M30 bolts with a strength grade of 8.8 as specimens for the early loosening detection trials. To capture images of the side of the bolt head, a bolt tensile test fixture was utilized to secure the bolt in place. A HiBestViewer high-speed camera with a resolution of 1280 × 1024 was employed to capture images of the bolt head’s side under various tensile loads. The distance between the camera lens and the head of the bolt was 60 cm, capable of supporting a sampling rate of up to 2420 frames per second (fps). To improve image clarity and contrast, the laboratory was equipped with additional lighting equipment. Tensile tests were conducted using a computer-controlled hydraulic tensile testing machine with a maximum tensile force of 1000 kN and an accuracy of 1% of the test force. The applied tensile force remained below the yield load of the bolts. The experimental setup is illustrated in Figure 6.

### 3.2. Experimental Details

Before the experiment, black and white matte paint was applied to create black and white scattering spots on the side of the bolt head. The tensile load applied to the bolt during the experiment is detailed in Table 1. To improve the accuracy of the experiment, each group of tests was repeated three times.

### 3.3. Study on the Effect of Light Intensity

In order to assess the impact of light intensity on detection accuracy and to explore the efficacy of the Retinex algorithm in detecting reduced accuracy under a lower light intensity, images of the bolt head were captured at various light intensities by adjusting the fill-in lamp’s light intensity while keeping the other experimental conditions constant. Figure 7 displays the images of the bolt head under three different light intensities.

### 3.4. Early Bolt Loosening Assessment Procedures

The flowchart of the early bolt loosening assessment based on the DIC method is shown in Figure 8. In the experiment, a relationship model between the mean strain on the side of the bolt head and the axial force is established through tensile testing. The process includes the following steps: First, an industrial camera is used to capture speckle images on the side of the bolt head. Then, the captured speckle images are preprocessed to ensure the quality of image information. Next, the DIC method is used to calculate the mean strain on the side of the bolt head region from the preprocessed images, and finally, a mathematical model of the relationship between the mean strain on the side of the bolt head and the axial force is constructed. In practical applications, the mean strain on the side of the bolt head region is first determined using the DIC method, and then the axial force acting on the bolt under working conditions is calculated based on the mathematical model of the relationship between the mean strain on the side of the bolt head and the axial force to determine whether there are signs of loosening in the bolt.

## 4. Results and Discussion

### 4.1. Results

Under moderate-light-intensity conditions, we plotted the relationship between mean strain and axial tension in the center region of the lateral side of the head of the bolt, and, based on the results of the three experiments for each group of bolts, we calculated the juxtaposed fitting curve of the mean strain and axial tension in this lateral side, as shown in Figure 9. This figure demonstrates that within a given range, the mean strain in this region exhibits a linear growth trend as the axial tensile force increases. Due to errors in the tensile testing machine at low load ranges and the application of a pre-tension load during the tensile test, the mean strain is not directly proportional to the axial tensile force. Therefore, a linear function is utilized to model the mean strain and axial tensile force, and the fitting results are presented in Table 2. It is evident that after linear fitting, the goodness of fit R^2^ exceeds 0.93, indicating a strong linear relationship between the mean strain in the central region of the bolt head side surface and the axial tensile force.

### 4.2. Experimental Results on Light Intensity Investigation

In environments with weak lighting, the quality of the image of the bolt head side can be affected, as shown in Figure 10a. In this case, the linearity between the mean strain of the central region of the bolt head side and the axial load decreases, with a goodness of fit R2 value less than 0.95, as shown in Figure 10a, Figure 11a and Table 3. To improve the image quality, the Retinex algorithm and multi-scale histogram stretching method are used to enhance the image of the bolt head side under weak lighting conditions, resulting in the processed image shown in Figure 10b,c, the fitting curve is shown in Figure 11b,c. The processed image is clearer compared to the original image, making feature recognition easier. The standard deviation of the intercept and slope of the mean strain in the lateral center region of the bolt head fitted to the axial tension was reduced by 10.56% and 10.32%, respectively, through the Retinex image enhancement process. This reduction improved linearity. Additionally, the standard deviation of the intercept and slope was enhanced by 15.40% and 15.49%, respectively, using the histogram stretching method. However, this enhancement did not significantly improve linearity. This indicates that the multi-scale Retinex algorithm can effectively mitigate the adverse effects of insufficient lighting on the quality of digital images of the bolt head side.

### 4.3. Method Validation

To assess the accuracy and practicality of this approach, a tensile testing machine was employed to subject bolts to axial tensile loads exceeding those specified in Table 1. The mean strain in the central region of the bolt head side was determined using the DIC method. Subsequently, by correlating the fitted mean strain with the axial force as shown in Table 2, the axial force on the bolt was approximated. This estimation was compared with the actual applied load to verify the accuracy and practicality of this approach.

Based on the linear fitting results of Table 2, the M30 bolt has the worst linear fit, and to verify the accuracy of the experiment, we chose the M30 bolt with the worst fit for subsequent experimental verification, with the applied loads of 15 and 80 kN, following the experimental details described in Section 3.2. The mean strains of the central region of the M30 bolt heads in two groups were measured. Substituting the juxtaposed fitted equations shown in Table 2, the predicted axial tensile force of the M30 bolts under this strain was calculated, as shown in Table 4. When applying 15 kN to the M30 bolts, the maximum predicted value error of this verification experiment according to the fitting equation in Table 2 was 1.57 kN, and the minimum was 0.38 kN, with relative mean errors of 10.47 and 2.53%, respectively. When applying 80 kN to the M30 bolts, the maximum predicted value error was 1.52 kN and the minimum was 0.01 kN, with relative mean errors of 1.90 and 0.01%, respectively. These results demonstrate the accuracy and practicality of the research method in calculating the axial tensile force of bolts.

## 5. Conclusions

Bolt connections are widely used in various industrial fields, and accurately and conveniently detecting the early loosening of bolts is a challenging problem. In response to the problem of high-pressure manifold leakage due to the loosening of bolts, this study proposes a new method for calculating the axial force of bolts based on DIC technology, achieving the non-contact detection of early bolt loosening. The main conclusions are as follows:(1)A research method for detecting bolt loosening based on DIC is proposed by using digital image processing techniques and analyzing small deformations on the object’s surface. To the best of the authors’ knowledge, this is the first application of the DIC method to detect strain on the side of the bolt head and establish a relationship model between this strain and the axial load to achieve the non-contact detection of early bolt loosening.(2)There is a good linear relationship between the mean strain in the central region of the side of the bolt head and the axial force of the bolt, allowing for the prediction of axial force based on this mean strain. In the verification experiment, the maximum relative mean error in predicting axial force based on this mean strain does not exceed 11%.(3)Retinex image enhancement processing reduced the standard deviation of the intercept and slope of the mean strain versus axial tension fit in the lateral center region of the bolt head by 10.56% and 10.32%, respectively, improving its linearity.

However, the proposed method still has some limitations. For instance, (1) corrosion and contamination can cause rust to form on the surface of the bolt, affecting the image quality and interfering with the accuracy of strain measurements; (2) the relationship between the elongation of the bolt rod and the preload force can be expressed by a functional equation, but there is no specific expression for the relationship between the strain on the bolt head side and the preload force; and (3) the method necessitates high-precision industrial cameras and image processing equipment, making it relatively costly. To enhance the practicality of this approach, future research can focus on the following aspects. (1) This method faces challenges in detecting bolt loosening under corrosive conditions, and future researchers can explore alternative methods to improve bolt loosening detection by integrating this method with other techniques, such as combining digital image technology with vibration detection technology to assess the bolt tightening state through comprehensive analysis of image features and vibration characteristics. (2) New, intelligent algorithms should be developed or incorporated to improve processing speeds and enhance automatic identification and analysis of the test region. (3) Since there are numerous bolt types and materials, the database of relationships between the strain on the side of bolt head and axial load can be expanded to provide data support for bolt loosening detection. (4) The high-speed camera used in this study is relatively expensive. New algorithms can be introduced to enable ordinary cameras to achieve the required detection accuracy, thereby improving the application prospects and practicability of this method in the field of bolt loosening detection.

## Figures and Tables

**Figure 1 sensors-24-05397-f001:**
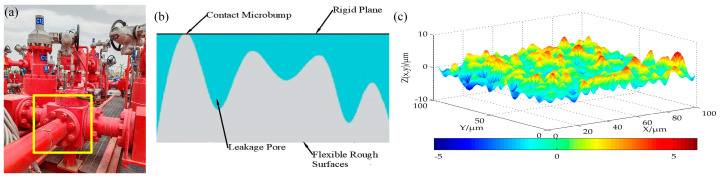
Simplified model of contact surfaces [22]. (**a**) High-pressure manifold flange bolts, (**b**) simplified contact model for rough surfaces, and (**c**) 3D rough surface modeling.

**Figure 2 sensors-24-05397-f002:**
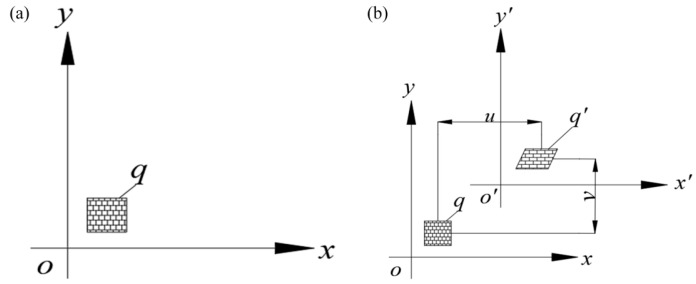
DIC technology schematic. (**a**) Relationships among subsets of the pre-deformation scatter pattern; (**b**) relationships among subsets of the post-deformation scatter pattern.

**Figure 3 sensors-24-05397-f003:**
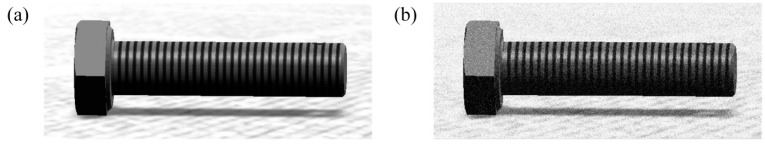
Imaging noise demonstration. (**a**) Original image; (**b**) image with added Gaussian noise.

**Figure 4 sensors-24-05397-f004:**
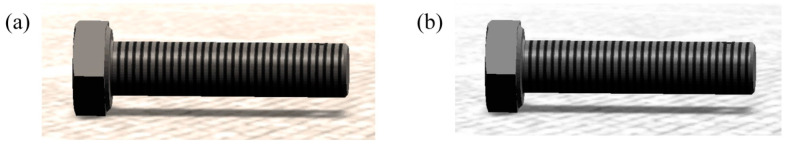
Greyscale image. (**a**) Original image; (**b**) greyscale image.

**Figure 5 sensors-24-05397-f005:**
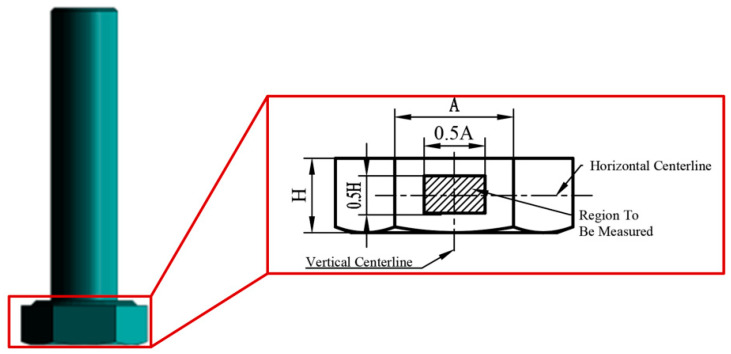
Strain calculation region on the side of the bolt head.

**Figure 6 sensors-24-05397-f006:**
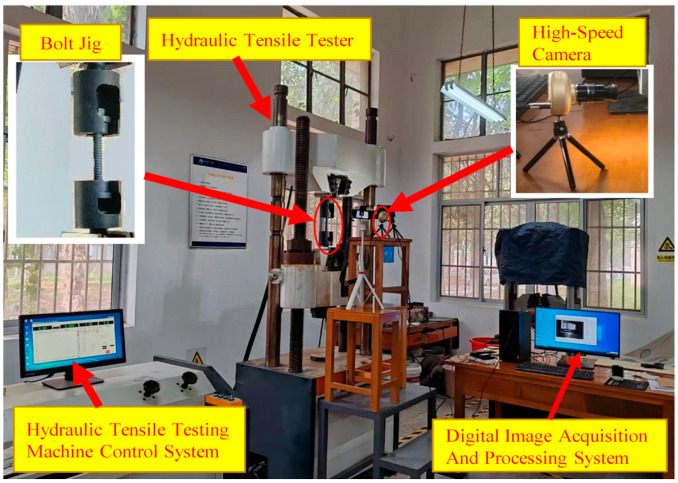
Experimental setup.

**Figure 7 sensors-24-05397-f007:**

Images of the bolt head’s side under different light intensities: (**a**) weak, (**b**) moderate, and (**c**) strong light.

**Figure 8 sensors-24-05397-f008:**
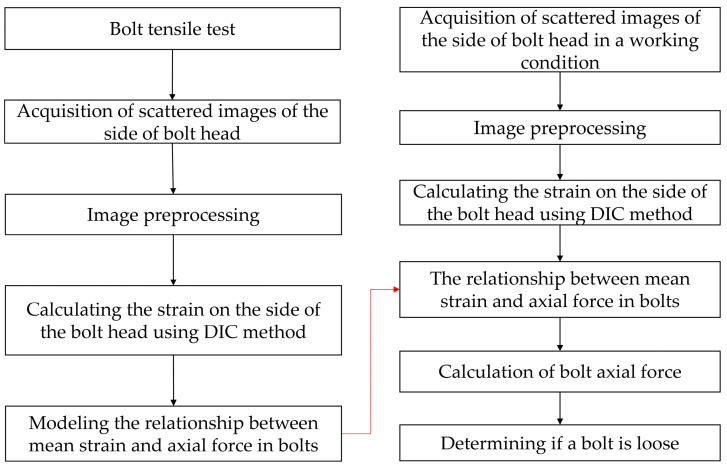
Research methodology flowchart.

**Figure 9 sensors-24-05397-f009:**
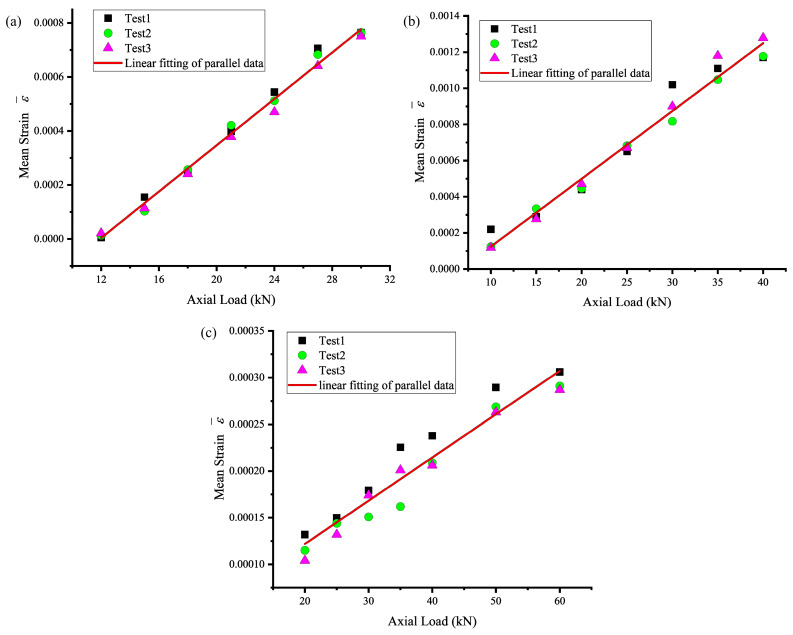
Relationship between the mean strain of the central region on the side of a bolt head and the axial load. (**a**) M20, (**b**) M24, and (**c**) M30.

**Figure 10 sensors-24-05397-f010:**
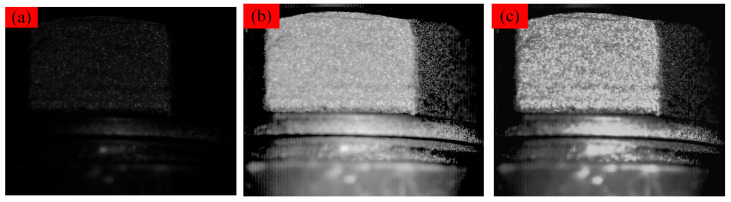
Image comparison. (**a**) Original image, (**b**) Retinex-enhanced image, and (**c**) histogram-stretch-enhanced image.

**Figure 11 sensors-24-05397-f011:**
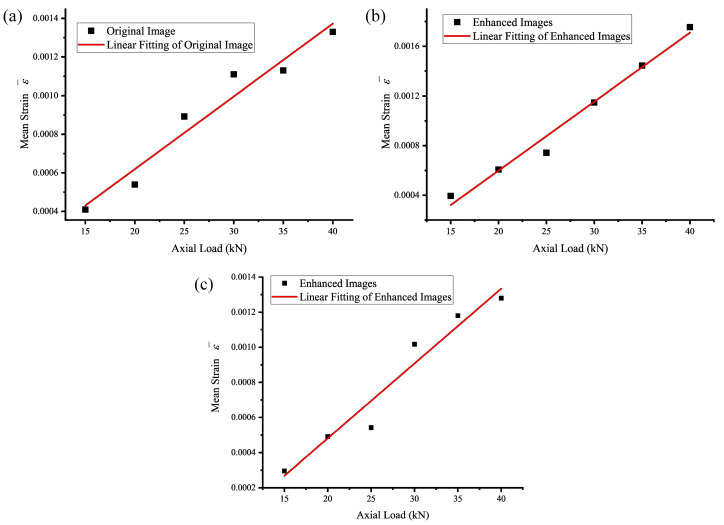
Relationship between the mean strain of the central region of the side of the bolt head under weak light conditions and after image enhancement with changes in axial load. (**a**) Weak light; (**b**) Retinex-enhanced image linear fitting; (**c**) histogram-stretch-enhanced image fit.

**Table 1 sensors-24-05397-t001:** The value of bolt tensile load.

Bolt Type	Load/kN						
M20	12	15	18	21	24	27	30
M22	10	15	20	25	30	35	40
M30	20	25	30	35	40	50	60

**Table 2 sensors-24-05397-t002:** Results of juxtaposed linear fitting results of mean strain in the central region of the bolt head side and axial load.

Bolt Type	Intercept (×10^−4^)	Standard Error of Intercept (×10^−5^)	Slope (×10^−5^)	Standard Error of Slope (×10^−6^)	Adjusted R^2^
M20	−5.11	2.05	4.29	0.94	0.99
M24	−2.51	3.67	3.75	1.36	0.97
M30	2.92	1.11	4.63	2.83	0.93

**Table 3 sensors-24-05397-t003:** Linear fitting results of mean strain and axial load in the central region of the bolt head side under weak light conditions and image enhancement.

Image Type	Intercept (×10^−4^)	Standard Error of Intercept (×10^−4^)	Slope (×10^−5^)	Standard Error of Slope (×10^−6^)	Adjusted R^2^
Original image	−1.35	1.23	3.77	4.26	0.94
Retinex-enhanced images	−5.12	1.10	5.56	3.82	0.98
Histogram-stretching-enhanced image	−3.72	1.42	4.26	4.92	0.94

**Table 4 sensors-24-05397-t004:** Predicted values and errors of bolt axial loads.

Bolt Type	M30	M30	M30	M30
Mean strain in the center region (×10^−4^)	1.06	0.97	4.07	4.00
Predicted values/kN	16.57	14.62	81.52	80.01
Actual applied load/kN	15.00	15.00	80.00	80.00
Error in prediction/kN	1.57	−0.38	1.52	0.01
Relative error of mean strain	10.47%	2.53%	1.90%	0.01%

## Data Availability

Data are contained within the article.

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
