# Peer review of "Early Bolt Loosening Detection Method Based on Digital Image Correlation"

_sensors, 2024, doi:10.3390/s24165397_

Round 1

Reviewer 1 Report

Comments and Suggestions for Authors

In this paper, a novel method for early assessment of bolt loosening based on digital image correlation is proposed. The method is interesting. However, there are still some issues that need improvement.

(1) The basis for determining the value should be provided in section 2.2.1. For example,  the size of the scattering to a range of 3 to 5 pixels, a scattering density of around 50%, and 33 scattering sizes.

(2) In Table2, the linear fitting is used for each test for different kinds of bolt. Why do not fit all the data from test 1~3 for each kind of bolt. 

(3) In section 4.2, the value of the parameter segam of  Gaussian filter function is not given and the influence of the parameter is not disscussed.

(4) Lack of bolt verification on actual or laboratory prototype nodes. 

(5) The parameter settings for the experiment should be supplemented, such as the distance of the camera, etc

(6) The relationship curves have all been established for M20, M24 and M30. Why only the M30 bolt is used in the experiment verification.

Author Response

Thank you very much for taking the time to review this manuscript. Your comments and suggestions guide our further study of the entire issue. According to your comments and suggestions, we have revised the manuscript and provided a point-by-point response to the reviewer's comments in the attached document.

Reviewer 2 Report

Comments and Suggestions for Authors

This paper developed a method for early assessing bolt-loosening based on digital image correlation. The bolt-loosening is quantified via an established relation model between average strain and axial force. Overall, this manuscript is a minor extension in the field, and the reviewer suggests the following revisions to improve the manuscript's quality.

1.       Some English sentences written in the manuscript are unclear enough, have errors (e.g., used tense, etc.). Thus, the reviewer recommend that the author must carefully revise those in the manuscript again.

2.       The introduction should be revised to make it clear that the scientific problems the work aimed at overcoming or the limitations existed. In this case, scientific problem could be summarized and make the contributions more clearly.

3.       Do the author think the proposed method can be applied to detect bolt-loosening of corroded bolts? and what are the issues?

4.       In conclusion chapter, besides summarizing the completed work and achievements, the authors should discuss more about their limitations/drawbacks of proposed method, should include suggestions for future studies aimed at addressing these limitations, and provide insight into the potential directions for further developments in this research.

Comments on the Quality of English Language

The english is good with minor errors

Author Response

(The authors gave the same response as above.)

Reviewer 3 Report

Comments and Suggestions for Authors

Article Reference: sensors-3096046-peer-review-v1

Detail Comments on the article “Early Bolt Loosening Detection Method Based on Digital Image Correlation

General Overview:

This article studied the detection technique of loosening bolts by adopting DIC. The work contains both theoretical & experimental parts. Though the DIC technique is nothing new in the area of monitoring and detection work but still this work may assist people those who are working in the same area. Similar information is useful but if a work doesn’t have enough novelty then it does not make sense to be published as a journal.

However, there are many similar or very close studies can be found in existing literatures. To avoid any misunderstanding, few are given below, the following articles are recommended to check:

Xie, W. G., Zhou, P., Chen, L. Y., Gu, G. Q., Wang, Y. Q., & Chen, Y. T. (2023). Non-Destructive Detection of High-Strength Wind Turbine Bolt Looseness Using Digital Image Correlation. Metrology and Measurement Systems, 30(4), 839–850. https://doi.org/10.24425/mms.2023.147948

Xie, C., Luo, J., Li, K., Yan, Z., Li, F., Jia, X., & Wang, Y. (2024). Image-Based Bolt-Loosening Detection Using a Checkerboard Perspective Correction Method. Sensors, 24(11), 3271. https://doi.org/10.3390/s24113271

Xie, W. G., Zhou, P., Chen, L. Y., Gu, G. Q., Wang, Y. Q., & Chen, Y. T. (2023). Non-Destructive Detection of High-Strength Wind Turbine Bolt Looseness Using Digital Image Correlation. Metrology and Measurement Systems, 30(4), 839–850. https://doi.org/10.24425/mms.2023.147948

Deng, L., Sa, Y., Li, X., Lv, M., Kou, S., & Gao, Z. (2024). A Vision-Based Bolt Looseness Detection Method for a Multi-Bolt Connection. Applied Sciences (Switzerland), 14(11). https://doi.org/10.3390/app14114385

Zhang, Y., Sun, X., Loh, K. J., Su, W., Xue, Z., & Zhao, X. (2020). Autonomous bolt loosening detection using deep learning. Structural Health Monitoring, 19(1), 105–122. https://doi.org/10.1177/1475921719837509

Gao, Z., He, X., Li, W., & Xiong, K. (2015). Digital Image Correlation-Based Bolt Preload Monitoring, https://papers.ssrn.com/sol3/papers.cfm?abstract_id=4495482

Comments on the Quality of English Language

N/A

Author Response

(The authors gave the same response as above.)

Reviewer 4 Report

Comments and Suggestions for Authors

The authors outline an early bolt loosening detection method based on digital image correlation. I think there are large adjustments needed prior to acceptance. I recommend major revision. 

1. DIC has been around for quite sometime, however, it is notoriously slow due to the dense facets of pixels that are correlated from frame to frame. The authors have not referenced any literature pointing to common computer vision techniques that are used to combat this.   

2. What is the sample rate of the camera? You state that it is high speed, but I did not see it mentioned.

3. To me, the authors are using Retinex as a way to boost the lightning in the scene. However, why wouldn't we just histogram stretch the image? That's far more efficient and reduces the chances of artifact generation / pixel saturation.

4. Overall, I think the "novelty" that is claimed in this paper isn't novel. The authors are using a pre-defined technique "DIC" and applying it to a test case. Bolt loosening has been studied using DIC in literature. I would encourage the authors to re-vist their literature review.

Author Response

(The authors gave the same response as above.)

Round 2

Reviewer 1 Report

Comments and Suggestions for Authors

There is no comments and suggestions

Reviewer 4 Report

Comments and Suggestions for Authors

Following the prior revision, I am comfortable recommending for publication